# Content of Selected Macro- and Microelements in the Liver of Free-Living Wild Boars (*Sus scrofa* L.) from Agricultural Areas and Health Risks Associated with Consumption of Liver

**DOI:** 10.3390/ani10091519

**Published:** 2020-08-27

**Authors:** Anna Kasprzyk, Janusz Kilar, Stanisław Chwil, Michał Rudaś

**Affiliations:** 1Department of Pig Breeding and Biotechnology, Institute of Animal Breeding and Biodiversity Conservation, University of Life Sciences in Lublin, 13 Akademicka, 20-950 Lublin, Poland; 2Institute of Agricultural and Forest Economy, Jan Grodek State University in Sanok, 21 Mickiewicza, 38-500 Sanok, Poland; janusz.kilar@wp.pl; 3Podkarpacki Agricultural Advisory Center in Boguchwala, 9 Suszyckich, 36-040 Boguchwala, Poland; 4Department of Chemistry, Faculty of Food Science and Biotechnology, University of Life Sciences in Lublin, 15 Akademicka, 20-950 Lublin, Poland; stanislaw.chwil@up.lublin.pl; 5Central Laboratory of Research, University of Life Sciences in Lublin, 30 D Głęboka, 20-612 Lublin, Poland; michal.rudas@up.lublin.pl

**Keywords:** wild boar, toxic elements, essential elements, health risk assessment, hazard index

## Abstract

**Simple Summary:**

Liver is a traditional dish and a basic ingredient of traditional dishes consumed in many countries worldwide. The analysis of trace elements and macro- and microelements in the liver of wild boars is important due to the insufficient amount of available scientific information in this field. The material comprised liver samples taken from 70 wild boars (*Sus scrofa* L.). The animals represented the following ranges of age: Up to one year (group I), from one to three years (group II), and over three to five years (group III). It was shown that the wild boar liver is a rich source of mineral compounds. The age was found to exert an effect on the concentration of most minerals. The sex significantly determined the content of iron, calcium, and cadmium. The results obtained in our study indicate that children should avoid a frequent intake of wild boar liver. In turn, this type of liver can be a valuable source of such elements as Fe, Zn, and Cu for adults. The permissible concentration of Cd in the liver was exceeded in five samples. We suggest that regular control of the content of trace elements in wild boar liver is extremely important and advisable for assessment of the level of consumer exposure. The toxic metal content can be determined to assess the degree of environmental pollution and for biomonitoring of contamination. The information included in this article may be useful for environment protection agencies and policy makers that design strategies for environmental protection and human health and safety regulations.

**Abstract:**

The aim of the study was to determine the levels of selected toxic and non-toxic elements in the liver of free-living wild boars from agricultural areas and to assess health risks associated with liver consumption. Samples were collected from 70 wild boars. The animals were divided into three age groups (group I up to one year, group II from one to three years, group III over three to five years). It was shown that wild boar liver is a rich source of mineral compounds (K, Fe, Mg, Ca, Zn, and Cu). The age was found to exert an effect on the concentration of most minerals. The sex significantly determined the content of Fe, Ca, and Cd. The maximum allowable level of Cd in the liver was exceeded in two and three samples from groups I and III, respectively. Therefore, regular monitoring of the content of this element in tissues of game animals is extremely important and advisable to assess the consumer exposure to this metal. From the point of view of human health, the estimation of the non-carcinogenic risk indicated that the intake of individual trace elements through the consumption of the liver was safe, whereas consumption of combined trace elements (only in the case of the consumption of the wild boar liver twice a week) suggested a potential health risk to children.

## 1. Introduction

A vast majority of metals contained in Earth’s crust is represented by trace elements, which are indispensable for the proper function of living organisms. Their presence in the environment is a result of natural geological processes [1]. However, the increasing industrial production and agricultural activity have led to a drastic increase in environmental contamination (air, plants, soil, and water), and consequently, exposure of living organisms (plants, animals, and humans) to a variety of toxic substances, including heavy metals [2,3,4,5]. Cadmium contained in soil persists on its surface for many years. It becomes a component of humus, where it is easily absorbed by plants. The bioaccumulation and phytoavailability index is 10 in the case of cadmium and below one in the case of another equally harmful heavy metal, i.e., lead [6]. Over time, potentially toxic trace elements (PTE) can reach levels that will pose a serious threat to ecological balance [6]. Additionally, heavy metals are a serious hazard to human health. Increasing concentrations of Pb and Cd have been recorded in soil, water, and sediments from mining and smelting areas in Europe [2,3,7,8,9]. Their direct sources are wastewater, sewage, industrial and municipal dusts and smoke, and exhaust gases. The processes of coal burning, including the so-called low emission, contribute substantially to the emission of metals into the atmosphere. Due to their small sizes, dusts emitted in this way can be transported with air circulation over long distances and deposited not only in the immediate vicinity of emitters. As reported by Šajn et al. [9], emissions from some types of factories can contaminate the environment at distances up to 152 km. Given the observations of PTE pollution over large areas, research methods based on biological detectors of its accumulation have become important in recent years. The use of indicator species can provide data for monitoring the quality and contamination of the biological environment [5,10]. Such animal species as fish, birds, and rodents have been used as bioindicators of environmental pollution. The wild boar has also been reported in the literature [4,11,12] as a good or even excellent indicator of the environmental contamination with toxic elements. This species is a suitable bioindicator due to its wide geographical distribution (occurrence in many countries of the globe), nutritional habits (carnivore and herbivore), a relatively long lifespan (up to 20–30 years), and the ease of sampling facilitated by regular culling [13]. 

Lead and cadmium are one of the most common contaminants with a tendency towards bioaccumulation by both plants and animals and inefficient processes of natural detoxification. When absorbed by the mammalian organism, Cd and Pb accumulate mainly in the liver and kidneys [10,14,15]. The tendencies towards accumulation of Cd and Pb and the lack of an effective mechanism for elimination thereof pose a potentially serious threat to animal and human health [14]. In humans, cadmium is a carcinogenic and mutagenic element [16]. The presence of these metals in the human and animal environment is a serious health-related and ecological problem. The degree of heavy metal accumulation in organs depends on the amount of ingested food and water, soil contamination, airborne concentration, and biological half-life, which is 10–30 years in the case of Cd and Pb [17]. Even in trace amounts, cadmium is regarded as a toxic element due to limitation of biotransformation processes, which results in slow excretion thereof and accumulation in living organisms. Therefore, the pathological effects of cadmium to organs and tissues strongly indicate the need for controlling the amount of cadmium in foodstuffs. Hence, it is indispensable to monitor cadmium concentrations in the environment, cities, and industrial regions and in areas that are not directly related to sources of pollution [12]. Toxic metals are absorbed from the diet relatively easily. Some potentially toxic trace elements, e.g., Co, Cu, Fe, Se, and Zn, are indispensable for living organisms. However, when ingested at higher than allowable concentrations, they may disturb the functions of enzymes and are responsible for the occurrence of many conditions, in particular diseases of the nervous and cardiovascular system or bone disorders in mammals [11,16,18].

Red meat is an important source of minerals in the human diet, as it provides highly bioavailable elements required for normal development and health [19]. Minerals contained in meat, in comparison with those present in plants, are more easily absorbed [20]. Macro- and microelements are the building material of bones, teeth, skin, and hair and are fundamental components for metabolic processes, maintenance of acid-base equilibrium, and regulation of water and electrolyte metabolism [21]. To date, no specific limits have been established in the European Union for toxic metals in game meat and offal, and the regulations regarding lead and cadmium are limited to the minimum risk levels (MRLs) listed in Reg. 629/2008 for pigs and other farmed species [22].

The concentration of elements in animal tissues can be influenced by such factors as animals’ age and sex, sampling season (hunting), and nutrition [5,10,12,23,24]. A significant impact on the content of heavy metals is undoubtedly exerted by the living habitat of game animals, which is much more closely related to the environment than that of farm animals [25]. Since bioaccumulation of toxic metals originating from anthropogenic pollution is becoming a potential threat to human health [26], systematic monitoring of the presence of heavy metals is necessary not only in industrialized areas but also in agricultural regions [12]. To date, there are few studies assessing the heavy metal pollution in the liver of wild boars induced by agricultural activities. There is limited information available to consumers about the mineral content in the liver of the wild boar. The novelty of this study is the assessment of the potential consumer health risk related to consumption of livers of wild boars foraging in agricultural areas.

The aim of the study was to determine the levels of selected toxic and non-toxic elements in the liver of free-living wild boars from agricultural areas of Poland and to assess health risks associated with liver consumption.

## 2. Materials and Methods

### 2.1. Feeding Grounds of the Wild Boar 

The wild boars lived in an agricultural region (90,270 km^2^) in the southeast covering the area of the following voivodeships: Lubelskie, podkarpackie, podlaskie, and mazowieckie. This is an area with 59% of arable land and 31% of forests. Plants cultivated in the fields include cereals (wheat, maize, barley, and triticale), potatoes, fodder plants, and vegetables. The forests are dominated by pine as well as birch, oak, spruce, beech, and alder. The area is a typical lowland and upland agricultural region covered predominantly by brown soils with some admixture of rendzinas and podzolic soils. The animals lived in forests, crop fields, and meadows away from heavy industry [10]. The region is characterized by a moderate continental climate. The average annual precipitation rate is 514 mm and the average annual air temperature is 9.5 °C. South-western winds prevail in the region.

Samples of the posterior segment of livers collected from 70 wild boars in collaboration with official local hunting authorities were the research material. No ethical committee permission was required, as the samples were collected post-mortem. The wild boars were culled by hunters as part of the hunting economy scheme and hunting limits (Journal of Laws 1995 No. 147 item 713; Bill of 13 October 1995, Hunting law as amended) in Poland in the hunting season of 2017. The wild boars originated from regular controlled culling approved by the Ministry of Environment and carried out to obtain meat for consumption.

### 2.2. Sample Preparation

Approximately 300 g of liver tissue were dissected from the left lobe of the liver. All samples were divided into three age groups (group I up to one year young—16 animals, group II from one to three years—28 animals, group III over three to five years old—26 animals) taking into account the animals’ sex (females *n* = 32; males *n* = 38). The age of the wild boars was determined on the basis of external appearance (colour, height, length of fang wear). The samples were packed separately into plastic bags, transported to the laboratory, and stored at −18 °C until analysis.

### 2.3. Elemental Analysis

The samples were thawed at 4 °C for 24 h. Thawed samples were ground using an analytical mill and ca. 0.5-g aliquots of the homogeneous mass were weighed on an analytical balance with an accuracy of 0.0001 g and placed in Teflon tubes. The aliquots were flooded with 10 cm^3^ HNO_3_ (Suprapur-Merck). Capped tubes were transferred to the mineralizer rotor. The samples were mineralized in a CEM Mars Xpress Matthews, NC, USA microwave oven at a temperature of 210 °C and a pressure of approximately 7 atmospheres. The transparent mineralizates were quantitatively transferred to 50 cm^3^ volumetric flasks and diluted with demineralized water (conductivity 0.055 µS/cm) to the marking. The content of mineral components (K, Ca, Mg, Fe, Zn, Mn, and Cu) was determined using the flame atomic absorption spectrometry technique (AAS—Varian SpectrAA 280 FS). The determination was carried out based on the calibration curve using background correction with a deuterium lamp. All elements were analyzed in the air/acetylene stoichiometric flame. Spectral Schinkel buffer solution was used for some elements, e.g., K, Ca, and Mg. To determine the concentration of Cd, Pb, As, and Ni, the solutions were analyzed on an inductively coupled plasma mass spectrometer (ICP Mass Spectrometer Varian MS-820 Belrose, Australia). The following isotopes of the analyzed elements were used: ^60^Ni, ^75^As, ^111^Cd, ^206^Pb, ^207^Pb, and ^208^Pb. Ultra Scientific standards with 99.999% purity and Merck Suprapure nitric (V) and hydrochloric acids were used in the analysis. Validation parameters used in the analysis are shown in Table 1. The quality control in the analysis was achieved by measurements of the blank sample, double sample, and certified reference material “NIST-1577c Bovine Liver”. Nitrogen was determined with the Kjeldahl method on a Kjeltec 2100 Hilleroed, Denmark Foss Tecator device in accordance with the PN-A-04018 standard [27]. Phosphorus was determined colorimetrically following the methodology proposed by Fiske and Subbarowa [28] with the use of the Cecil 2011 (Cambridge, UK) apparatus. The analyses were performed in triplicate. The results were expressed in mg·kg^−1^ of meat weight. The research was conducted in the certified Central of Laboratory of Agroecology of the University of Life Sciences in Lublin.

### 2.4. Nutritional Assumptions Used for Assessment of the Dietary Intake of Elements and Potential Consumer Health Risk (HQ) and Hazard Index (HI)

The consumers were divided into three groups according to the frequency of consuming liver: Those consuming wild boar liver frequently (90 times a year), those consuming liver periodically (12 times a year), and those consuming liver occasionally (two times a year). A portion of liver was assumed to be 138.4 g for an adult and 111.2 g for a child. The potential health risk (HQ) for consumers was assessed based on the concentrations of selected elements contained in the liver. 

The risk values were calculated using the formula:HQ = EDI/RfD(1)
where EDI is the estimated daily intake (mg/kg b.w./day) and RfD is the reference dose [29].
(2)$$\mathrm{EDI}=\frac{C\;\times \;EF\;\times \;ED\;\times \;LC}{BW\;\times \;T}:1000\;\left[\mathrm{mg}/\mathrm{kg}/\mathrm{day}\right]$$
where C is the element content (μg/g), EF is the exposure frequency (days/year), ED is the duration of exposure (70 years for adults, six years for children), LC is the liver consumption (g), BW is the average body weight (70 kg for adults, 20 kg for children), and T is the average exposure time (365∙ED).

The hazard index (HI) was calculated as the sum of HQ values [29]. When the HQ value is less than 1, the exposed population is unlikely to experience any adverse health hazard. When the HQ exceeds 1, there might be concerns for potential noncancerous effects.

### 2.5. Statistical Analysis

The analyses were performed using the STATISTICA 13.1 software Kraków, Poland for analysis of data. The normality was assessed using the Kolmogorov-Smirnov test, and Levene’s homogeneity of variance test was applied to examine the equality of variances. To determine the effect of the age and sex (as well as their interaction) on the analyzed traits, two-way analysis of variance with Tukey’s test was used at a significance level of *p* ˂ 0.05 and *p* ˂ 0.01. Basic statistical calculations were performed for each variable with determination of the mean, standard error, geometric mean (GM), and the minimum and maximum values.

## 3. Results

The content of heavy metals is presented in Table 2. There were statistically significant differences (*p* ≤ 0.01) in the lead content in the livers between groups I and II. No statistically significant differences were observed in the content of Cd between the analyzed groups; however, higher values of this element were noted in the livers of the over three-year-old wild boars. Furthermore, the maximum allowable level of this metal in the liver (0.5 mg/kg) was exceeded in two and three samples from groups I and III, respectively.

The highest levels of the macroelements analyzed in the wild boar livers were recorded in the case of K, Ca, and Mg (Table 3). The content of potassium in group II was statistically significantly different from that in groups I and III (*p* ≤ 0.01). The livers from group III were characterized by the highest Mg concentration. Significant differences (*p* ≤ 0.05) in the concentration of this element were also observed between groups I and II. No statistically significant differences were noted in the case of the Ca content, but there was a tendency towards lower levels of the element in the liver samples from group III. There was only a slight variation in the level of N, and its mean content in the livers of the wild boars was 3610 mg·kg^−1^. Statistically significant differences (*p* ≤ 0.01) were noted in the phosphorus concentration between group II and the other groups.

In the case of microelements (Table 4), there were statistically significant differences (*p* ≤ 0.01) in the content of Fe between groups II and III. Substantially lower values of Zn and Cu were recorded in the livers from group I, compared with group II. The Ni content in all the groups was below the limit of quantification. The Mn content was statistically significantly higher (*p* ≤ 0.01) in group I in comparison with groups II and III.

The statistical analysis of the results (Table 2, Table 3 and Table 4) demonstrated that the animals’ sex had a significant effect on the content of Cd, Ca, and Fe. Livers sampled from males were characterized by a significantly higher level of Cd and Fe. In turn, livers taken from females exhibited a higher level of Ca. There was no effect of the sex on the content of other minerals, although there was a tendency towards a higher concentration of Zn in the livers of males. The analyses revealed an interaction between the age and sex with respect to the content of potassium, copper, and manganese.

The calculation of EDI for the three adopted scenarios and the HQ and HI in children and adults are presented in Table 5, Table 6 and Table 7. The highest HQ values were observed for Fe, Cd, Cu, and Zn in the consumed liver of groups I, II, and III. However, in all the scenarios, the HQ was found to be lower than 1. In the frequent consumption scenario, the hazard index (HI) values were higher than 1 for children. In the liver of wild boar, the highest HI values were recorded for Fe, Cd, Cu, and Zn: 40%, 27%, 14%, and 10%, respectively. In the case of adults, all HI values were below 1, both for the occasional and weekly consumption of the liver.

## 4. Discussion

A higher concentration of Cd was noted in the present investigations in the group of the youngest wild boars and over three year old animals. The concentration of this element in the former group of animals may be a result of higher intestinal absorption and the underdeveloped blood−brain barrier [30]. Halamić and Miko [31] found that the presence of cadmium in 83% of samples of kidney tissue at a concentration more than 10 times higher than that in the soil supports the tendency of cadmium accumulation in the body already in the first year of life. It has been proved that the younger the organism is, the greater its ability to absorb Cd is, and there is a positive correlation between cadmium concentrations and animals’ age [12,13]. The cadmium concentrations in the present study are in agreement with those reported by Rudy [10] and Gasparik et al. [32], who analyzed the content of this element in wild boar livers. The content of the metal exceeded the highest allowable level (0.5 mg·kg^−1^) in two samples from group I and three samples from group III. This may have been associated with the individual tendency to absorb and accumulate cadmium. As reported by Rudy [11], the maximum (0.5 mg·kg^−1^) permissible level of cadmium in the liver was exceeded only in two samples in the group of over three year old animals. Any quantitative changes in the environment are reflected in animal tissues but, as suggested by Rudy [10], the sensitivity of these “living sensors” is highly variable, even within the same species. Our data on the Cd content in the wild boar livers differ from the results published by Chiari et al. [24], who reported a Cd concentration of 0.52–0.64 mg·kg^−1^ in the livers of wild boars living in a mountainous environment in northern Italy. Slightly higher values were reported by Chiari et al. [24] in investigations on livers of wild boars from one of the districts in Italy and by Mandas [33], who analyzed the concentration of Cd in the livers of wild boars from Sardinia. As demonstrated by Szkoda et al. [1], particularly high concentrations of this metal (3.095 mg·kg^−1^ of fresh weight) were reported from Upper Silesia. A significantly higher Cd level was found in the males than in the females in the present study, which may be related to their diet. In comparison with females, wild boar males consume greater amounts of protein and are characterized by higher feed intake. Furthermore, invertebrates (i.e., snails) have been found to provide important links in transferring heavy metals from plant to carnivores [34].

A comparison of the present results with the current values specified by the EU recommendations, i.e., the allowable cadmium content in the liver of 0.5 mg/kg, allows a conclusion that the values demonstrated in this research are lower than those defined in the regulations [22]. Since the amount of cadmium in the wild boar livers does not exceed the maximum allowable level, these parts can be recommended for human consumption, although there is a general recommendation for consumers that ingestion of offal from game animals should be avoided [12]. In our opinion, lead poses a considerable threat to human health only at a high concentration, even at infrequent consumption of game meat. Nevertheless, there is a need to control venison in order to protect consumers’ health and undertake effective steps to eliminate this problem.

The average Pb content in this study was 0.21 mg·kg^−1^; therefore, the level of this element was lower than the EU minimum risk for swine offal, i.e., 0.5 mg·kg^−1^ ww. [22]. The mean contents detected in the liver were similar to those reported by Mandas [29] in wild boars aged 3–5 years from Sardinia, wild boars assessed by Rudy [10], and wild boars aged 1–5 years from central Poland analyzed by Długaszek and Kopczyński [35]. Pilarczyk et al. [29] detected an average Pb concentration of 0.702 mg/g in the liver of wild boars from western Ukraine, which was three times as high as that found in this study. A higher Pb concentration was recorded by Amici et al. [23] in wild boars living in six areas of Viterbo Province and by Živkov Balos et al. [5], who analyzed livers of wild boars originating from 16 regions of Serbia. In the case of age, Chiari et al. [24] did not find a difference between wild boars below and above 1 year of age, as in the present study. In wild boar habitats located near an industrial complex, Martelli [36] detected a substantial level of lead in the liver, kidneys, and muscles. Similar results were reported by Szkoda et al. [1], who analyzed livers of wild boars originating from Bogatynia and Upper Silesia, and by Durkalec et al. [37], who examined livers of wild boars living in an area regarded as the most toxic metal contaminated part of the Upper Silesia region. In their analyses of the content of toxic elements in wild boars from a mining area of the Ciudad Real Province in Spain, Taggart et al. [38] found an average Pb level in the liver of 1.675 mg·kg^−1^ d.w. In turn, the highest concentration of Pb, i.e., on average 2.28 mg·kg^−1^, was observed by Kryński et al. [39] in the livers of wild boars from south−western Poland.

The Pb concentrations detected in animal tissues indicate that this element is present in various components of the environment and is capable of bioaccumulation in the trophic chain. In 2010, on the request of the European Commission, the Scientific Panel on Contaminants in the Food Chain of the European Food Safety Authority [40] analyzed the results of toxicological studies conducted in recent years and new data on food contamination with lead. They concluded that the current value of provisional tolerable weekly intake (PTWI) of lead specified at 25 µg/kg b.w., i.e., 3.57 µg Pb/kg b.w./day, should be replaced by lower doses determining the BMDL (Benchmark Dose Lower Confidence Limit). The lowest doses are associated with induction of a specific effect on the human organism. Permissible values of the index were set at the following levels: BMDL_01_ for children (neurotoxic effect) 0.50 µg·kg^−1^ b.w./day as well as BMDL_10_ 0.63 µg·kg^−1^ b.w./day (nephrotoxic effect) and BMDL_01_ 1.50 µg·kg^−1^ b.w./day (cardiovascular disorders) for adults [40].

In terms of the As content, a similar concentration was detected by Taggart et al. [38] and Kucharczak and Motyl [25], who assessed the content of this element in the livers of wild boars living in industrial areas, agricultural regions, and urban agglomerations. Referring to the levels of this element recorded in the literature, it was found that the wild boars examined in the present study had much lower levels of As than those in Ukraine [29]. In an analysis of the content of trace elements in the wild boar livers, Rudy [10] reported contents of arsenic lower than 0.001 mg·kg^−1^. The European Union regulations do not specify the maximum level of As in meat and liver; therefore, the interpretation of our result in this aspect is somewhat difficult.

Potassium is one of the macronutrients. It is an important element in water−electrolyte metabolism and acid-base homeostasis in the organism. The variability of the K concentration in the present study was in the range of 1820−3410 mg·kg^−1^. Similar levels of this element were reported by Sales and Kotrba [41] in meat from wild boars from Slovakia. The results of the Ca concentration coincide with those observed by Długaszek and Kopczyński [35]. Ca plays an important role in the process of organism growth. The mean Mg content in the livers of the analyzed wild boar groups was similar to that recorded by Długaszek and Kopczyński [35]. Substantially higher values of the Mg concentration in the range of 810−1060 mg·kg^−1^ were observed by Roślewska et al. [42]. These differences may be associated with the dissimilarities in the animals’ habitats, differences in the feed base, and the different methodology of determination of the component.

The investigations reported by Babicz et al. [19] show that edible slaughter by products are the main store of Fe. Importantly, Fe contained in offal is heme iron, the absorption of which is several-fold higher than the absorption of non-heme iron contained in other raw materials and food products. The high levels of these elements are associated with their high oxidative metabolism in the cells [43]. The wild boar liver is a rich source of iron; its content in the present study was, on average, 429.53 mg·kg^−1^ at SD = 181.18. As shown by Pilarczyk et al. [29], the Fe content in wild boar liver was highly variable and ranged from 82.40 to 301.00 mg·kg^−1^. A substantially lower variability in the concentration of this component was reported by Skobrák et al. [44].

The livers sampled from the males in the present study exhibited a higher level of Fe and a lower concentration of Ca, compared with gilts. It can be assumed that the higher Fe levels are associated with the higher activity and increased oxidative metabolism in males [19]. The mean zinc content in the wild boar livers analyzed in this study was similar to the value reported by Roślewska et al. [42] and higher than that recorded by Skobrák et al. [44]. A high level of zinc in the dose is not dangerous to the organism, as this element is easily excreted in feces [45]. With regard to the Cu content, similar results were reported by Roślewska et al. [42], who analyzed wild boar meat (from males and females). In investigations of the Cu concentration in wild boar meat, Długaszek and Kopczyński [18] found lower values and a high variability of this component (from 0.15 to 1.79 mg·kg^−1^). The content of Mn in the present study was the highest in the livers of the youngest wild boars, which is probably associated with individual tendencies to absorb and accumulate metals. As demonstrated by Długaszek and Kopczyński [18], the content of Mn in wild boar meat varied widely in the range from 0.08 to 1.39 mg·kg^−1^. The differences in the content of some minerals (K, Mg, P, Cd, Zn, Cu, and Mn) in relation to animals’ age and sex may be related to several factors. As reported by Ping et al. [46], the bio-concentration of metals depends on the differences in metal uptake, storage, and regulatory mechanisms within species. 

In this study, the values of HQ of individual metals in the liver were below 1, which indicates that the frequent intake of these metals is unlikely to cause adverse health effects in adult consumers. However, a frequent intake of the liver studied may exert adverse health effects on children due to the content of Fe, Cd, Cu, and Zn. The zinc concentration was within the allowable daily doses of this element for an adult human (10−40 mg/day, after World Health Organization). As specified by the Food and Agriculture Organization of the United Nations/World Health Organization recommendations, the tolerable consumption of cadmium by an adult human is approx. 0.4−0.5 mg/week, and the allowable dose is 60−70 μg per day [6,47]. Dietary levels of protein, zinc, copper, calcium, and iron compounds exert a considerable impact on the uptake of cadmium from the gastrointestinal tract. Low levels of these elements in food increases absorption of cadmium from the gastrointestinal tract and accumulation in the organism. Increased amounts of zinc in food reduce the intensity of cadmium absorption from the gastrointestinal tract. Therefore, the likelihood of appearance of symptoms of cadmium intoxication immediately after consuming wild boar liver is small, even if the concentration of this metal exceeds the allowable levels, as the digestibility of the element is approx. 10% [47]. The Cd level in the liver of wild boars from groups I and III only in two and three samples, respectively, slightly exceeded the maximum permissible concentration of this element in foodstuffs. However, this metal has the ability to bioaccumulate, e.g., in the liver and kidneys [10,14]. Increased cadmium content in food has a remarkably negative impact on the health of children, as it reduces the immune resistance of young organisms. Moreover, cadmium is one of the most serious factors increasing the risk of cancer diseases in children [47]. The offal is an excellent source of highly available Na, Ca, and trace elements (mainly Fe, Mn, and Cu) [20]. Bioavailability of iron is greatest when it is in the heme form, and meat is the primary source of this form. As reported by Babicz et al. [19], meat-derived iron is characterized by approximately 20−30% availability. It was found that the HI values associated with the consumption of liver of wild boar were 2.5 fold higher for children than for adults. Similar values were also reported by Pilarczyk et al. [29]. Kicińska et al. [15] reported that in scenarios with daily and weekly consumption of animal liver, the final HI calculated for children was approx. 3.5 times higher than for adults. Kicińska et al. [15] reported a potential risk to health associated with the consumption of liver from game animals (HI > 1). In contrast, Lazarus et al. [48] indicated that consumption of game meat and liver in Croatia did not pose a health risk to consumers, regardless of the exposure scenario. Nevertheless, the authors suggested that children should avoid eating game liver, because a high percentage of samples were found to exceed the permitted limits for Cd and Pb.

## 5. Conclusions

Liver from wild boar can be an important source of minerals in the diet of hunting communities. The age was found to exert an effect on the concentration of Pb, K, Mg, P, Fe, Mn, Cu, and Zn. The Fe, Cu, and Zn contents increased with age, whereas the Mn content decreased significantly. The livers from young wild boars contained the highest amounts of Pb and P. In turn, the livers of animals from group III accumulated the highest levels of K, Fe, Mg, Zn, and Cu. The investigations have confirmed increasing accumulation of Fe in the wild boar livers with age. The sex significantly determined the content of Fe, Ca, and Cd. The liver of the female boars contained less Fe and Cd but more Ca. The higher content of Fe in the livers from the male than female animals may be associated with the higher muscle weight and hemoglobin content in males. In the group of the analyzed toxic elements (Pb, Cd, As), the highest and lowest levels were determined for Cd and As, respectively. The concentration of Pb and As in the analyzed livers does not raise concern. These values are generally lower than those recorded in other European countries. The liver Cd concentration in 12.5% (up to one year old wild boars) and 11.5% (over three year old animals) of the liver samples were above the European Union Maximum Residue Levels (MRL) established for offal destined for human consumption. The concentration of this element in the young wild boars may be a result of the higher intake thereof with sow’s milk. In the livers of older wild boars, it may be associated with migration of the animals into non−agricultural areas characterized by a higher cadmium concentration. However, the mean Cd levels were in the range of physiological concentrations and did not exceed the permissible temporal limits specified for the liver of farm animals. However, regular control of the content of this element in game animal tissues is extremely important and advisable for assessment of the level of consumer exposure to this metal. Additionally, further research is required to identify the source of pollution in order to protect the health of both animals and humans. This research shows that the younger the organism is, the greater its ability to absorb Pb and Cd is. The average Pb content in this study was lower than the EU minimum risk for swine offal. This confirms the satisfactory status of the agricultural environment in the area of wild boar foraging (crop fields, forests). Children should avoid a frequent intake of liver from wild boar. Based on the calculated HI values, it seems recommendable that consumption of the liver by children should be limited to one such a meal monthly. Adults can consume wild boar liver even twice a week.

## Figures and Tables

**Table 1 animals-10-01519-t001:** Results of the analysis of certified reference materials and proficiency tests.

Element	LOQ (mg·kg^−1^)	Certified Value (mg·kg^−1^)	Analyzed Value (mg·kg^−1^)	Recovery (%)
Ca	56	131	129.17	99
Mg	36	620.42	651.87	105
Fe	8.6	197.94	190.53	96
Mn	2.2	10.46	10.27	97
Cu	2.6	275.20	274.50	99
Zn	0.9	181.10	102.88	106
K	40	10.23	9.45	94
Cd	0.8	0.097	0.097	100
Pb	1.3	0.063	0,069	110
As	0.9	0.019	0.019	97
Ni	1.1	0.045	0.047	106

LOQ: Limit of quantification.

**Table 2 animals-10-01519-t002:** Concentration of trace elements in wild boar liver (mg·kg^−1^ wet weight).

Element	Parameter	Group I	Group II	Group III	Female	Male	Effects of Age	Effects of Sex	Interaction A × S
Pb	Means ± SE	0.2636 ^A^±0.116	0.1630 ^B^ ± 0.108	0.2015 ± 0.107	0.195 ± 0.103	0.194 ± 0.116	**	ns	ns
	GM	0.2420	0.1423	0.1764	0.1740	0.1641			
	Range	0.122−0.520	0.05–0.470	0.07−0.38	0.090−0.470	0.048−0.520			
Cd	Means ± SE	0.4405 ± 0.226	0.4227 ± 0.178	0.4485 ± 0.284	0.3802 ^b^ ± 0.229	0.4863 ^a^ ± 0.200	ns	*	ns
	GM	0.3917	0.3561	0.3870	0.2942	0.4509			
	Range	0.02−1.00	0.17−0.23	0.22−1.03	0.020−1.00	0.22−1.03			
As	Means± SE	0.0469 ± 0.012	0.0404 ± 0.016	0.0313 ± 0.012	0.0379 ± 0.013	0.0435 ± 0.016	ns	ns	ns
	GM	0.0453	0.0375	0.0200	0.0356	0.0404			
	Range	0.0315–0.0659	0.0186−0.0791	0.0120−0.0560	0.0186−0.0692	0.0120−0.0791			

A: Age; S: Sex; SE: Standard error; GM: Geometric mean; * means between the sexes in the same row with different letters are significantly different—a, b, *p* ˂ 0.05; ** means between the groups in the same row with different letters are significantly different—A, B, *p* ˂ 0.01; ns: Not significant.

**Table 3 animals-10-01519-t003:** Concentration of macroelements in wild boar liver (mg·kg ^−1^ w.w.).

Element	Parameter	Group I	Group II	Group III	Female	Male	Effects of Age	Effects of Sex	Interaction A × S
––	Means ± SE	2565 ^A^ ± 390.36	2260 ^B^ ± 249.09	2709 ^A^ ± 429.77	2432 ± 296.73	2374 ± 410.81	**	ns	**
	GM	2535.59	2246.59	2680.03	2415.46	2340.85			
	Range	1820−3110	1821−2638	2120−3410	1978−3110	1820−3410			
Ca	Means ± SE	181.23 ± 104.34	249.76 ± 127.39	122.94 ± 83.87	232.96 ^a^ ± 117.42	164.73 ^b^ ± 114.34	ns	*	ns
	GM	152.72	187.47	101.96	199.87	129.98			
	Range	65.5−380	32.8−458	43.2−269	43.2−458	32.8−385			
Mg	Means ± SE	191.63 ^a^ ± 17.07	174.06 ^b B^ ± 22.12	209.63 ^A^ ± 23.90	183.25 ± 14.58	183.69 ± 30.84	**	ns	ns
	GM	190.92	172.36	208.52	182.67	180.82			
	Range	169−220	102−200	188−261	150−208	102−261			
N	Means ± SE	3610 ± 317	3537 ± 388	3618 ± 416	3607 ± 436	3558 ± 313	ns	ns	ns
	GM	3647	3516	3597	3581	3545			
	Range	3163−4200	2660−4230	2970−4240	2660−4231	2960−4240			
P	Means ± SE	1276 ^A^ ± 59	1066 ^B^ ± 164	1276 ^A^ ± 175	1138 ± 188	1160 ± 168	**	ns	ns
	GM	1274	1052	1260	1120	1147			
	Range	1150−1350	700−1350	1000−1560	750−1560	701−1400			

SE: Standard error; GM: Geometric mean; * Means between the sexes in the same row with different letters are significantly different—a, b *p* ˂ 0.05; ** means in the same row with different letters are significantly different—A, B *p* ˂ 0.01; ns: Not significant.

**Table 4 animals-10-01519-t004:** Concentration of microelements in wild boar liver (mg·kg ^−1^ w.w.).

Element	Parameter	Group I	Group II	Group III	Female	Male	Effects of Age	Effect of Sex	Interaction A × S
Fe	Means ± SE	408.51 ± 135.69	344.33 ^B^ ± 98.27	535.75 ^A^ ± 89.81	331.35 ^B^ ± 88.97	435.64 ^A^ ± 193.35	**	**	ns
	GM	390.28	331.73	476.56	321.46	404.70			
	Range	259−696	234−553	286−1063	234−553	238−1063			
Cu	Means ± SE	6.79 ^b^ ± 0.99	8.55 ^a^ ± 2.52	8.82 ^a^ ± 2.94	8.54 ± 2.64	7.75 ± 2.12	*	ns	*
	GM	6.73	8.21	8.42	8.19	7.48			
	Range	5.09−8.44	4.37−14.90	5.60−13.50	5.87−14.90	4.37−13.50			
Zn	Means ± SE	50.24 ^b^ ± 7.66	61.10 ^a^ ± 14.98	63.20 ^a^ ± 12.13	54.73 ± 14.39	62.77 ± 12.00	*	ns	ns
	GM	49.70	59.20	62.18	61.62	53.01			
	Range	40.3−62.4	31.6−89.00	44.3−85.20	44.3−83.34	31.6−89.00			
Ni	Means ± SE	<LOQ = 0.1	<LOQ = 0.1	<LOQ = 0.1	<LOQ = 0.1	<LOQ = 0.1	ns	ns	ns
	GM	0.099	0.095	0.091	0.097	0.093			
	Range	<LOQ = 0.1	<LOQ = 0.1	<LOQ = 0.1	0.08−0.10	0.07−0.10			
Mn	Means ± SE	3.300 ^A^ ± 0.505	2.711 ^B^ ± 0.536	2.487 ^B^ ± 0.432	2.847 ± 0.724	2.831 ± 0.441	**	ns	*
	GM	3.26	2.66	2.45	2.76	2.79			
	Range	2.7−4.3	2−4	1.9−3.1	1.9−4.3	2−4.3			

SE: Standard error; GM: Geometric mean; LOQ: Limit of quantification; * Means in the same row with different letters are significantly different—a, b *p* ˂ 0.05; ** means in the same row with different letters are significantly different—A, B *p* ˂ 0.01.

**Table 5 animals-10-01519-t005:** Association of the estimated daily intake of elements, health risk (HQ), and hazard index (HI) with consumption of liver of the wild boar from group I.

Element	Estimated Daily Intake (EDI, mg/kg b.w.)						Health Risk (HQ) for Consumers					
	Frequent Consumption (90 times/year)		Periodic Consumption (12 times/year)		Occasional Consumption (two times/year)		Frequent Consumption (90 times/year)		Periodic Consumption (12 times/year)		Occasional Consumption (two times/year)	
	Children	Adult	Children	Adult	Children	Adult	Children	Adult	Children	Adult	Children	Adult
Fe	0.46526	0.19026	0.06203	0.02536	0.01033	0.00422	0.66466	0.27181	0.06203	0.03624	0.01477	0.00604
Cu	0.00802	0.00328	0.00106	0.00043	0.00017	7.29 × 10^−5^	0.20049	0.08198	0.02673	0.01093	0.00445	0.00182
Zn	0.05924	0.02422	0.00789	0.00043	0.00131	0.00053	0.19474	0.08075	0.02633	0.01076	0.00438	0.00179
Ni	0.00011	0.00004	1.58 × 10^−5^	6.45 × 10^−6^	2.63 × 10^−6^	1.07 × 10^−6^	0.00591	0.00241	0.00078	0.00032	0.00013	0.00005
Mn	0.00388	0.00158	0.00051	0.00021	8.64 × 10^−5^	0.00003	0.02776	0.01135	0.00370	0.00151	0.00061	0.00025
As	5.4 × 10^−5^	0.00002	7.2 × 10^−6^	0.000002	2.20 × 10^−6^	4.91 × 10^−7^	0.00385	0.00157	0.00051	0.00021	8.57 × 10^−5^	0.00003
Pb	0.00028	0.00011	3.85 × 10^−5^	0.00001	6.41 × 10^−6^	2.62 × 10^−6^	0.08242	0.03370	0.01099	0.00449	0.00183	0.00074
Cd	0.00046	0.00019	0.000060	0.00002	0.00001	4.24 × 10^−6^	0.46696	0.19095	0.06226	0.02546	0.01037	0.00424
HI							1.64955	0.67457	0.21994	0.08994	0.03665	0.00869

**Table 6 animals-10-01519-t006:** Association of the estimated daily intake of elements, health risk (HQ), and hazard index (HI) with consumption of liver of the wild boar from group II.

Element	Estimated Daily Intake (EDI, mg/kg b.w.)						Health Risk (HQ) for Consumers					
	Frequent Consumption (90 times/year)		Periodic Consumption (12 times/year)		Occasional Consumption (two times/year)		Frequent Consumption (90 times/year)		Periodic Consumption (12 times/year)		Occasional Consumption (two times/year)	
	Children	Adult	Children	Adult	Children	Adult	Children	Adult	Children	Adult	Children	Adult
Fe	0.39545	0.16171	0.05272	0.02156	0.00878	0.00035	0.56493	0.00926	0.07532	0.03080	0.01255	0.00513
Cu	0.00978	0.00400	0.00130	0.00053	0.00021	8.89 × 10^−5^	0.24456	0.10001	0.03260	0.01333	0.00543	0.00222
Zn	0.07057	0.02886	0.00941	0.00384	0.00156	0.00064	0.23521	0.09620	0.03136	0.01282	0.00522	0.00213
Ni	0.00011	4.63 × 10^−5^	1.51 × 10^−5^	6.18 × 10^−6^	2.52 × 10^−6^	1.03 × 10^−6^	0.00566	0.00231	0.00075	0.00030	0.00012	5.14 × 10^−5^
Mn	0.00317	0.00129	0.00042	0.00017	7.05 × 10^−5^	2.88 × 10^−5^	0.02265	0.00926	0.00302	0.00123	0.00050	0.00020
As	4.47 × 10^−5^	1.82 × 10^−5^	5.96 × 10^−5^	2.44 × 10^−5^	9.93 × 10^−7^	4.06 × 10^−7^	0.00319	0.00130	0.00042	0.00017	7.10 × 10^−5^	2.90 × 10^−5^
Pb	0.00017	6.94 × 10^−5^	2.26 × 10^−5^	9.25 × 10^−6^	3.77 × 10^−6^	1.54 × 10^−6^	0.04846	0.01982	0.00646	0.00262	0.00107	0.00044
Cd	0.00042	0.00017	5.66 × 10^−5^	2.31 × 10^−5^	9.43 × 10^−6^	3.86 × 10^−6^	0.42452	0.17360	0.05660	0.02314	0.00943	0.00385
HI							1.54923	0.63354	0.20656	0.08447	0.03442	0.01407

**Table 7 animals-10-01519-t007:** Association of the estimated daily intake of elements, health risk (HQ), and hazard index (HI) with consumption of liver of the wild boar from group III.

Element	Estimated Daily Intake (EDI, mg/kg b.w.)						Health Risk (HQ) for Consumers					
	Frequent Consumption (90 times/year)		Periodic Consumption (12 times/year)		Occasional Consumption (two times/year)		Frequent Consumption (90 times/year)		Periodic Consumption (12 times/year)		Occasional Consumption (two times/year)	
	Children	Adult	Children	Adult	Children	Adult	Children	Adult	Children	Adult	Children	Adult
Fe	0.56693	0.23184	0.07559	0.03091	0.01259	0.00515	0.80990	0.33102	0.10798	0.04416	0.01799	0.00736
Cu	0.01003	0.00410	0.00133	0.00054	0.00022	9.12 × 10^−5^	0.25092	0.10261	0.03345	0.01368	0.00557	0.00228
Zn	0.07412	0.03031	0.00988	0.00404	0.00164	0.00067	0.24707	0.10103	0.03294	0.01347	0.00549	0.00224
Ni	0.00010	4.44 × 10^−5^	1.45 × 10^−5^	5.92 × 10^−6^	2.41 × 10^−6^	9.87 × 10^−7^	0.00543	0.00222	0.00072	0.00029	0.00012	4.93 × 10^−5^
Mn	0.00291	0.00119	0.00038	0.00015	6.49 × 10^−5^	2.65 × 10^−5^	0.02086	0.00853	0.00278	0.00113	0.00046	0.00019
As	2.38 × 10^−5^	9.57 × 10^−6^	3.18 × 10^−6^	1.30 × 10^−6^	5.30 × 10^−7^	2.17 × 10^−7^	0.00170	0.00069	0.00022	9.29 × 10^−5^	3.78 × 10^−5^	1.55 × 10^−5^
Pb	0.00021	8.60 × 10^−5^	2.80 × 10^−5^	1.15 × 10^−5^	4.67 × 10^−6^	1.91 × 10^−6^	0.06008	0.02457	0.00801	0.00327	0.00133	0.00054
Cd	0.000461	0.00018	6.15 × 10^−5^	2.52 × 10^−5^	1.03 × 10^−5^	4.19 × 10^−6^	0.46135	0.18866	0.06151	0.02515	0.01025	0.00419
HI							1.85734	0.75954	0.24764	0.10127	0.04127	0.01687

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
