# Peer review of "Content of Selected Macro- and Microelements in the Liver of Free-Living Wild Boars (Sus scrofa L.) from Agricultural Areas and Health Risks Associated with Consumption of Liver"

_animals, 2020, doi:10.3390/ani10091519_

Round 1
Reviewer 1 Report
Accept in present form
Author Response
Dear Reviewer,
We would like to thank the Reviewer for careful and thorough reading of this manuscript and for the thoughtful comments and constructive suggestions, which help to improve the quality of this manuscript. We corrected the whole manuscript very carefully according to comments and suggestions of Reviewer. Below we describe our responses point by point to each comment. Changes are marked by red colour.
Response:
(e.g. line 110) = page 3; L 167-168 (verified version): The sentence now reads:
“The novelty of this study is the assessment of the potential consumer health risk related to consumption of livers of wild boars foraging in agricultural areas.”
Yours faithfully,
Authors
Reviewer 2 Report
no
Author Response
Dear Reviewer,
We would like to thank the Reviewer for careful and thorough reading of this manuscript and for the thoughtful comments and constructive suggestions, which help to improve the quality of this manuscript. We corrected the whole manuscript very carefully according to comments and suggestions of Reviewer. Below we describe our responses point by point to each comment. Changes are marked by red colour.
Response:
Page 3; L 190-191:
“The wild boars originated from regular controlled culling approved by the Ministry of Environment and carried out to obtain meat for consumption.”
Yours faithfully,
Authors
This manuscript is a resubmission of an earlier submission. The following is a list of the peer review reports and author responses from that submission.
Round 1
Reviewer 1 Report
Manuscript entitled „Content of Selected Heavy Metals and Macro- and Microelements in the Liver of Free-Living Wild Boars (Sus Scrofa L.) from the Agricultural Areas and the Health Risks Associated with Liver Consumption” presents the results of the determination of selected elements in livers of wild boars living in Poland.
The paper contains some significant errors of a substantive, editorial and formal nature, which should be taken into account at the stage of improving the manuscript.
General remarks:
- Assessment of the content of elements in wet weights of samples, with water content up to 70%, raises serious objections as to errors in the correctness of the determination.
- The presented (in fact, their complete absence !!!) assumptions and parameters involved with assessing the health risk with must be definitely supplemented.
Specific remarks:
- The title of paper: I suggest removing the term "heavy metals" due to the fact that they are also trace elements, some of which the authors include in other groups (macro- and microelements).
Keywords: adequate to the content presented in the manuscript.
Abstract: adequate to the content presented in the manuscript.
Introduction:
- Please standardize the font throughout the manuscript, nowadays texts written with different sizes and fonts are very bad to read.
- lines 87-89: It is not true. Please see the work by Kicińska et al. (2019), Bakowska et al. (2016), Burger (2007), there were described the content of elements in the livers of many species of animals. As well, in paper of Kicińska et al. (2019) the health risk associated with the consumption of these livers has been calculated.
Materials and methods:
2.1. Sample preparation
- This section contains a long description of where the samples were taken. The authors writing "region" do not mention what is this region? Please complete this information. Please distinguish a new subsection. It is terribly m now.
- “The animals lived in a rather uncontaminated areas of forests and meadows, far away from heavy industry.” This whole sentence is an assumption, not information based on scientific facts. Please base your description on scientific data or avoid such expressions.
2.2. Elemental analysis
- The methodology of decomposing a sample of wet tissues (containing large amounts of water, approx. 70%) is in my opinion not precise when determining the content of trace elements !!! Please take a look: Adrian, In. J., & Stevens, M. L. (1979). Wet versus dry weight for heavy metal toxicity determination in duck liver. Journal of Wildlife Diseases, 15, 125–126.
2.3. Nutritional assumptions used to assess dietary intake of elements and potential health risk (HQ), hazard 142 index (HI) for consumers
- The basic parameters adopted for the HQing calculation are missing. It is absolutely necessary to expand this chapter in the methodology. It is currently insufficient.
- Line 122: In the methodology, the authors state that they made the determination of Na content, while there are no results in any table? why are several terms introduced: macro-, microelements, mineral components? heavy metals? why? Please provide a definition of these concepts, as they are used for various groups of elements.
Results:
- Tab. 2. As is not heavy metal, it is a metalloid!!!
- Why Fe, Mn, Cu and Zn were included among the microelements? since their contents are higher than that of some macroelements (e.g. N and P).
- The concentrations given for P and N in Table 3 are surprisingly low for the livers.
- It is not possible to assess the entire subchapter related to health risk, because the authors did not provide basic assumptions by which their results can be verified.
Discussion:
- Line 153 and 155: The authors incorrectly cited the name "Kicicińska" - and it is probably an author from their country.
- A lot of the information contained in the discussion has little to do with it, and should be carried over for introduction.
Conclusion:
The conclusions are quite laconic. I propose to expand them.
Regards
Author Response
We would like to thank the Reviewers for careful and thorough reading of this manuscript and for the thoughtful comments and constructive suggestions, which help to improve the quality of this manuscript. We corrected the whole manuscript very carefully according to comments and suggestions of Reviewers. Below we describe our responses point by point to each comment. Changes are marked by red colour.
Reviewer 1:
Comments: The paper contains some significant errors of a substantive, editorial and formal nature, which should be taken into account at the stage of improving the manuscript.
General remarks:
Assessment of the content of elements in wet weights of samples, with water content up to 70%, raises serious objections as to errors in the correctness of the determination.
The presented (in fact, their complete absence !!!) assumptions and parameters involved with assessing the health risk with must be definitely supplemented.
Response:
1. A standard procedure with a specified level of uncertainty was used for weighing the samples. The uncertainty was taken into account during the determinations of the samples.
2. As suggested by the reviewer the assumptions and parameters involved with assessing the health risk, we have supplemented; as shown in the revised manuscript; Page 5, L 239-246 (verified version).
Reviewer 1:
Specific remarks:
The title of paper: I suggest removing the term "heavy metals" due to the fact that they are also trace elements, some of which the authors include in other groups (macro- and microelements).
Response: We appreciate the positive feedback from the reviewer.
Page 1; L 2-4; As suggested by the reviewer, we have removed the term "heavy metals" from the title. The sentence now reads: “Content of Selected Macro- and Microelements in the Liver of Free-Living Wild Boars (Sus Scrofa L.) from the Agricultural Areas and the Health Risks Associated with Liver Consumption”.
Reviewer 1:
Introduction:
4. Please standardize the font throughout the manuscript, nowadays texts written with different sizes and fonts are very bad to read.
Response: The correction has been made, as shown in the revised manuscript.
Reviewer 1:
5. lines 87-89: It is not true. Please see the work by Kicińska et al. (2019), Bakowska et al. (2016), Burger (2007), there were described the content of elements in the livers of many species of animals. As well, in paper of Kicińska et al. (2019) the health risk associated with the consumption of these livers has been calculated.
Response: We have reviewed carefully the entire literature and Kicińska et al. (2019) reported mineral levels measured in the liver of a relatively small number of animals (N=10 wild boars, including 6 males and 4 females) from Poland. Most material was collected from farm animals (n = 22 samples), including 11 chickens, 4 rabbits, 2 cows, 2 goats, 1 pig, 1 duck, and 1 turkey; hence our claim that the wild boar population has been insufficiently investigated in terms of the mineral composition in the liver.
The aim of Bakowska et al. (2016) study was to evaluate the level of lead (Pb) in the livers and kidneys of free-living animals from Poland, with regard to the differences in tissue Pb content between the species [roe deer (Capreolus capreolus L.), red deer (Cervus elaphus L.), and wild boar].
Burger (2007) describes general gender-related differences in the levels of metals and other pollutants in fishes, birds, and some mammals, but there is no information about this issue in the case of the wild boar.
In our opinion, there is limited information available to consumers about the mineral content in liver.
lines 87-89 = page 3; L 118-121 (verified version): The sentence has been corrected to read: “There is limited information available to consumers about the mineral content in the liver of wild boar”. The novelty of this study is the analysis of the effect of consumption of liver from this animal on human health”.
Reviewer 1:
Materials and methods:
2.1. Sample preparation
6. This section contains a long description of where the samples were taken. The authors writing "region" do not mention what is this region? Please complete this information. Please distinguish a new subsection. It is terribly m now.
Response: The suggested correction has been made. Distinguished a new subsection.
Page 3, L 126-144:
“2.1. Feeding grounds of wild boar”
“Wild boars lived in a south-eastern agricultural region (90 270 km2) covering the area of the following voivodeships: lubelskie, podkarpackie, podlaskie and mazowieckie.” This is a area with 59% of arable land and 31% of forests….”…
“Samples of the posterior segment of livers collected from 70 wild boars in collaboration with authorised local hunting …”… “The wild boars were culled by hunters as part of the hunting economy scheme and hunting limits (Journal of Laws 1995 No. 147 item 713; Bill of October 13, 1995, Hunting law as amended) in Poland in the hunting season of 2017.”
“2.2. Sample preparation”
“Approximately 300 g of liver tissue was dissected from the left lobe of the liver. …”
Reviewer 1:
7. “The animals lived in a rather uncontaminated areas of forests and meadows, far away from heavy industry.” This whole sentence is an assumption, not information based on scientific facts. Please base your description on scientific data or avoid such expressions.
Response: The suggested correction has been made. lines 104-105 = Page 3, L 134 (verified version): The sentence now reads: “The animals lived in a areas of forests, crop fields and meadows, far away from heavy industry (Rudy 2010) [11].”
Reviewer 1:
2.2. Elemental analysis
8. The methodology of decomposing a sample of wet tissues (containing large amounts of water, approx. 70%) is in my opinion not precise when determining the content of trace elements !!! Please take a look: Adrian, In. J., & Stevens, M. L. (1979). Wet versus dry weight for heavy metal toxicity determination in duck liver. Journal of Wildlife Diseases, 15, 125–126.
Response: A standard procedure with a specified level of uncertainty was used for weighing the samples. The uncertainty was taken into account during the determinations of the samples.
The maximum levels for certain contaminants in foodstuffs specified by European Union regulations (European Commission, 2008. Commission Regulation (EC) No 629/2008 of 2 July 2008 setting maximum levels for certain contaminants in foodstuffs. Off. J. Eur. Union L 173/6, 3.7.2008.), which we refer to, are given in mg/kg wet weight. The wet weight for determination of heavy metal toxicity in liver was taken into account in a majority of investigations cited by us, e.g.: studies by Bakowska et al. 2016, Pilarczyk et al. 2020, Roślewska et al. 2016, Durcalec et al. 2015, Chiari et al. 2015, Crnić et al. 2015, Lazarus 2014, Ihiedioha et al. 2014, Długaszek and Kopczyński 2013, Amici et al. 212, Gasparik et al. 2012, Kucharczak and Motyl 2012, Rudy 2010, Ping et al. 2009. For this reason, we have specified the values in mg/kg wet weight (w.w.) to be able to relate the presented results to the reference literature.
Reviewer 1:
2.3. Nutritional assumptions used to assess dietary intake of elements and potential health risk (HQ), hazard index (HI) for consumers.
9. The basic parameters adopted for the HQing calculation are missing. It is absolutely necessary to expand this chapter in the methodology. It is currently insufficient.
Response: As suggested by the reviewer, we supplemented this chapter, as shown in the revised manuscript. Were used parameters which have been adopted by Pilarczyk et al. (2020).
line 150 = Page 5, L 239-246 (verified version):
“…where: EDI - estimated daily intake (mg/kg bw/day), RfD - reference dose [29].
where: C – element content (mg/g), EF – exposure frequency (days/hear), ED – exposure duration (70 years for adults, 6 years for children), LC – liver consumption (g), BW - average body weight (70 kg for adults, 20 kg for children), T - average exposure time (365∙ED).”
“Hazard index (HI) was calculated as the sum of HQ values [29]. If the HQ value is less than 1, the exposed population is unlikely to experience any adverse health hazard. If the HQ exceeds one, there might be concerns for potential noncancerous effects.”
Reviewer 1:
10. Line 122: In the methodology, the authors state that they made the determination of Na content, while there are no results in any table? why are several terms introduced: macro-, microelements, mineral components? heavy metals? why? Please provide a definition of these concepts, as they are used for various groups of elements.
Response: We did not determine of Na. This was our mistake. line 122 = Page 4, L 206, 211, 225 (verified version): Na has been deleted from the methodology.
Reviewer 1:
Results:
10. Tab. 2. As is not heavy metal, it is a metalloid!!!
Response: The suggested correction has been made. Page 6: The head of the table now reads:
“Table 2. Concentration of trace elements in wild boar liver (mg·kg-1 wet weight).”
Reviewer 1:
11. Why Fe, Mn, Cu and Zn were included among the microelements? since their contents are higher than that of some macroelements (e.g. N and P).
Response: The lower content of N and P than Fe was related to the fact that the content of these elements in tab. 2 is shown in % (g/100g) instead of mg/kg as stated in the head of the table. On the other hand, an increased concentration of Fe and Mn (over 1000 mg/kg) may occur in plant and animal organisms due to the large amount of the assimilable form of this component in soil.
Reviewer 1:
12. The concentrations given for P and N in Table 3 are surprisingly low for the livers.
Response: The contents of these elements in Tab. 2 are shown in % (g/100g) instead of mg/kg as stated in the head of the table. The concentrations for N and P have been corrected following the comments.
“Table 3. Concentration of macroelements in wild boar liver (mg·kg-1 w.w.).
Item
Parameter
Group I
Group II
Group III
Female
Male
K
Means ±SE
2565A±390.36
2260B±249.09
2709A±429.77
2432±296.73
2374±410.81
GM
2535.59
2246.59
2680.03
2415.46
2340.85
Range
1820-3110
1821-2638
2120-3410
1978-3110
1820-3410
Ca
Means ± SE
181.23±104.34
249.76±127.39
122.94±83.87
232.96a±117.42
164.73b±114.34
GM
152.72
187.47
101.96
199.87
129.98
Range
65.5-380
32.8-458
43.2-269
43.2-458
32.8-385
Mg
Means ± SE
191.63a±17.07
174.06bB±22.12
209.63A±23.90
183.25±14.58
183.69±30.84
GM
190.92
172.36
208.52
182.67
180.82
Range
169-220
102-200
188-261
150-208
102-261
N
Means ± SE
3610±317
3537±388
3618±416
3607±436
3558±313
GM
3647
3516
3597
3581
3545
Range
3163-4200
2660-4230
2970-4240
2660-4231
2960-4240
P
Means ± SE
1276A±59
1066B±164
1.276A±175
1138±188
1160±168
GM
1274
1052
1260
1120
1147
Range
1150-1350
700-1350
1000-1560
750-1560
701-1400
**Means in the same row with different letters are significantly different: A, B P˂ 0.01; ns = not significant
Reviewer 1:
13. It is not possible to assess the entire subchapter related to health risk, because the authors did not provide basic assumptions by which their results can be verified.
Response: The suggested correction has been made. Missing information was completed, as shown in the revised manuscript; page 5, L 239-246.
Reviewer 1:
Discussion:
14. Line 153 and 155: The authors incorrectly cited the name "Kicicińska" - and it is probably an author from their country.
Response: The correction has been made; lines 153 and 155 = Page 15, L 245 and 247 (verified version): „…Kicińska et al. [14] …”
Reviewer 1:
15. A lot of the information contained in the discussion has little to do with it, and should be carried over for introduction.
Response: Reviewer suggestion has been taken into consideration. We have reviewed carefully the entire manuscript and have carried over for introduction, as shown in the revised manuscript.
Reviewer 1:
Conclusion:
The conclusions are quite laconic. I propose to expand them.
Response: The suggested correction has been made. Page 15-16, L 255-291: The conclusion now reads:
“Liver from wild boar can be an important source of minerals in the diet of hunting communities. The age was found to exert an effect on the concentration of Pb, K, Mg, P, Fe, Mn, Cu, and Zn. The Fe, Cu, and Zn contents increased with age, while the Mn content decreased significantly. The livers from young wild boars contained the highest amounts of Pb and P. In turn, the livers of animals from group III accumulated the highest levels of K, Fe, Mg, Zn, and Cu. The investigations have confirmed the increase in the accumulation of Fe in the wild boar livers with age. The sex significantly determined the content of Fe, Ca, and Cd. The liver of the female boars contained less Fe and Cd but more Ca. The higher content of Fe in the livers from the male than female animals may be associated with the higher muscle weight and hemoglobin content in males. In the group of the analyzed toxic elements (Pb, Cd, As), the highest and lowest levels were determined for Cd and As, respectively. These values are generally lower than those recorded in other European countries. The liver Cd concentration in 12.5% (up to 1-year-old wild boars) and 11.5% (over 3-year-old animals) of the liver samples were above the European Union Maximum Residue Levels (MRL) established for offal destined for human consumption. The concentration of this element in the young wild boars may be a result of the higher intake thereof with sow's milk. In the livers of older wild boars, it may be associated with migration of the animals into non-agricultural areas characterized by a higher cadmium concentration. However, the mean Cd levels were in the range of physiological concentrations and did not exceed the permissible temporal limits specified for the liver of farm animals. However, regular control of the content of this element in game animal tissues is extremely important and advisable for assessment of the level of consumer exposure to this metal. Additionally, further research is required to identify the source of pollution in order to protect the health of both animals and humans. This research shows that the younger the organism is, the greater its ability to absorb Pb and Cd is. The average Pb content in this study was lower than the EU minimum risk for swine offal. This confirms the satisfactory status of the agricultural environment in an area of wild boar foraging (crop fields, forests). Children should avoid frequent intake of liver from wild boar. Based on the calculated HI values, it seems recommendable that consumption of the liver by children be limited to one such a meal monthly. Adults can consume wild boar liver even twice a week.”
Yours faithfully,
Authors

Reviewer 2 Report
The idea of the study is good. I have the following questions and suggestions.
1. I do not agree with the phrase: No ethical committee permission
was required as the samples were collected post-mortem.
The study had to be approved by a research and / or ethics committee. I suggest that it be evaluated by a committee.
2. what was the death of wild boars?
3. According to the division in the three groups. which group is young wild boars and which old age?
4. please be clearer, in table 2, ** Means in the same row with different letters are significantly different: A, B P˂ 0.01
You mean group I vs. group II vs. group III?
5. In table 2, if a statistic of t studens and anova is used, significant differences are found between the groups.
6.- In table 3, I suggest putting a charge on each ion. Thank you
7. In Table 3, I suggest putting the units of each ion.
8. I suggest to put the ions correctly, for example Ca2+ or Fe2+ or Fe3+. Thank you
Author Response
We would like to thank the Reviewers for careful and thorough reading of this manuscript and for the thoughtful comments and constructive suggestions, which help to improve the quality of this manuscript. We corrected the whole manuscript very carefully according to comments and suggestions of Reviewers. Below we describe our responses point by point to each comment. Changes are marked by red colour.
Reviewer 2:
Comments 1. I do not agree with the phrase: No ethical committee permission
was required as the samples were collected post-mortem.
The study had to be approved by a research and / or ethics committee. I suggest that it be evaluated by a committee.
Response: In accordance with the law (Journal of Laws 1995 No. 147 item 713; Bill of October 13, 1995, Hunting law as amended), the wild boar is designated as a big game animal by the Ministry of Environment in Poland. Hunting is carried out for the purposes of selection. The so-called reductive culling is carried out. The liver was taken from culled wild boars. Samples were collected during a regular hunting season from state hunting grounds in Poland. The wild boars were culled by hunters as part of the hunting economy scheme and hunting limits. Therefore, no ethical committee permission was required, as the samples were collected post-mortem. In total, samples from 70 animals (wild boar) that had been shot during the hunting season by the huntsmen within officially determined hunting limits were collected. The research was performed on biological material (liver) derived from wild boars were culled by hunters; after evaluation, meat is standardly intended for consumption. Therefore, our research does not require the approval of Animal Experimentation Committee.
Reviewer 2:
Comments 2. what was the death of wild boars?
Response: The wild boars were culled with the use of a caliber 308 W rifle and brass-shelled bullets. A shot at the heart caused immediate death of the animal.
Reviewer 2:
Comments 3. According to the division in the three groups. which group is young wild boars and which old age?
Response: The suggested correction has been made. Page 3, L 146-148; The sentence now reads:
“All samples were divided into 3 age groups (group I up to 1 year – young – 16 animals, group II from 1 to 3 years – 28 animals, group III over 3 to 5 years – old – 26 animals) taking into account the animals' sex (females n=32; males n=38).”
Reviewer 2:
Comments 4 please be clearer, in table 2, ** Means in the same row with different letters are significantly different: A, B P˂ 0.01
You mean group I vs. group II vs. group III?
Response: The correction has been made. Page 6.The sentences now reads:
“*Means between the sexes in the same row with different letters are significantly different: a, b P˂ 0.05; **Means between the group in the same row with different letters are significantly different: A, B P˂ 0.01; ns = not significant.”
Reviewer 2:
Comments 5 In table 2, if a statistic of t studens and anova is used, significant differences are found between the groups.
Response: The sentence now reads:
**Means between the group in the same row with different letters are significantly different: A, B P˂ 0.01; ns = not significant.”
Reviewer 2:
Comments 6. In table 3, I suggest putting a charge on each ion.
Response: In the methodology, we adopted the determination of the general forms of the elements. Therefore, expressing them in the ionic form would require a separate laboratory analysis of some of them. Therefore, we must keep in the presentation of the results in the tables in the atomic rather than ionic form.
Reviewer 2:
Comments 7. In Table 3, I suggest putting the units of each ion.
Response: The units are specified in the head of Table 3. The N and P contents were shown in g/100 g, which has been replaced by mg/kg.
The contents of these elements in Tab. 2 are shown in % (g/100g) instead of mg/kg as stated in the head of the table. The concentrations for N and P have been corrected following the comments.
Reviewer 2:
Comments 8. I suggest to put the ions correctly, for example Ca2+ or Fe2+ or Fe3+.
Response: See comment 6.
Yours faithfully,
Authors
